# Nanostructured and Spiky Gold Shell Growth on Magnetic Particles for SERS Applications

**DOI:** 10.3390/nano10112136

**Published:** 2020-10-27

**Authors:** Erin E. Bedford, Christophe Méthivier, Claire-Marie Pradier, Frank Gu, Souhir Boujday

**Affiliations:** 1Laboratoire de Réactivité de Surface (LRS), CNRS, UMR 7197, Sorbonne Université, 4 place Jussieu, F-75005 Paris, France; eebedford@gmail.com (E.E.B.); christophe.methivier@sorbonne-universite.fr (C.M.); claire-marie.pradier@upmc.fr (C.-M.P.); 2Department of Chemical Engineering and Waterloo Institute for Nanotechnology, University of Waterloo, 200 University Ave. W., Waterloo, ON N2L 3G1, Canada; frank.gu@uwaterloo.ca

**Keywords:** plasmonic nanoparticles, magnetic nanoparticles, nanostructures, SERS, XPS, surface functionalization, seed-growth, colloidal stability, magnetoplasmonic

## Abstract

Multifunctional micro- and nanoparticles have potential uses in advanced detection methods, such as the combined separation and detection of biomolecules. Combining multiple tasks is possible but requires the specific tailoring of these particles during synthesis or further functionalization. Here, we synthesized nanostructured gold shells on magnetic particle cores and demonstrated the use of them in surface-enhanced Raman scattering (SERS). To grow the gold shells, gold seeds were bound to silica-coated iron oxide aggregate particles. We explored different functional groups on the surface to achieve different interactions with gold seeds. Then, we used an aqueous cetyltrimethylammonium bromide (CTAB)-based strategy to grow the seeds into spikes. We investigated the influence of the surface chemistry on seed attachment and on further growth of spikes. We also explored different experimental conditions to achieve either spiky or bumpy plasmonic structures on the particles. We demonstrated that the particles showed SERS enhancement of a model Raman probe molecule, 2-mercaptopyrimidine, on the order of 10^4^. We also investigated the impact of gold shell morphology—spiky or bumpy—on SERS enhancements and on particle stability over time. We found that spiky shells lead to greater enhancements, however their high aspect ratio structures are less stable and morphological changes occur more quickly than observed with bumpy shells.

## 1. Introduction

Functional micro- and nanoparticles are key components in advanced detection methods, including applications in disease diagnosis, environmental monitoring, and as tools in research labs [1,2,3]. Multifunctionality, i.e., when a single particle can perform multiple tasks, is increasingly desired in these pursuits and can be achieved by specifically tailoring these particles during their synthesis or further functionalization [4,5]. Many multifunctional particles incorporate plasmonic nanostructures, often gold- or silver-based, due to their unique optical features [6,7,8,9]. These properties are exploited by certain optical surface-sensitive characterization methods to exalt optical signals detected upon biomolecule binding through electromagnetic enhancement or plasmonic effects [10,11]. These include surface plasmon resonance (SPR) [12,13,14], localized surface plasmon resonance (LSPR) [15,16,17,18], surface enhanced infrared absorption spectroscopy (SEIRAS) [19,20], metal enhanced fluorescence (MEF) [21,22], sum frequency generation [23,24] and surface enhanced Raman scattering (SERS) [25,26]. SERS has been heavily investigated as a transduction technique for biosensing due to the fingerprint specificity of Raman spectroscopy and the extremely large enhancements seen on certain substrates (on the order of 10^4^–10^7^) [26]. SERS sensing can be achieved on nanostructured planar substrates or on colloidal particles, where the surface to bulk ratio is higher than planar substrates and, therefore, the binding of an analyte favored, unconstrained by 2D geometrical limitations. However, unlike planar surfaces, where high control over the area of interaction and detection is possible, colloidal suspensions of plasmonic nanoparticles can be more difficult to handle and/or separate from the analyzed milieu. By providing them with functionality that allows spatial control, as is commonly done by making the particles magnetic, we can overcome this challenge and allow for applications, such as combined separation and sensing, to be achieved [27].

Magnetic particles for the sensitive and selective separation of biomolecules are unbeatable in terms of simplicity and low-cost, as only particles and a magnet are needed [28]. Among them, superparamagnetic particles are optimal as their magnetic properties are exhibited only under the influence of a magnetic field. Therefore, they remain dispersed in the solution to be analyzed, permitting analyte binding to their surface, yet can be easily separated by applying a magnet [29,30]. A diameter lower than 30 nm is required for Fe_3_O_4_ particles to be superparamagnetic [31,32], but such a small size slows down and complicates the separation of the particles, especially once coated [33,34]. Using controlled aggregates (150–300 nm) of superparamagnetic nanoparticles (~20 nm) increases the total magnetic moment while still maintaining the superparamagnetic properties of the particles, thus leading to easily magnetically separable particles [35]. 

Several strategies have been applied to assemble magnetic and plasmonic particles, including electrostatic interactions [36], microfluidic approaches [37], and most frequently seed-growth processes. The latter approach can be easily used to form gold shells on magnetic cores to achieve spherical magnetic core/plasmonic shell particle structures [38,39,40,41]. However, the degree of SERS signal enhancement in such a configuration is limited as the greatest signal amplifications are observed at specific electromagnetic hot-spots [25,42], i.e., sharp tips [43] and nanogaps [44,45], which only exist in defect areas of individual solid shells [46]. Anisotropic particles can provide a larger SERS signal [47,48,49,50], since anisotropic nanostructures can result in greater electromagnetic enhancement on a single particle, rather than depending on the intermittent nanogaps that arise when particles are drifting free in solution [51]. As a result, the growth of anisotropic gold nanostructures on the magnetic core is ideal for SERS application [42,52]. In addition to providing the electromagnetic enhancement, leading to the SERS effect, surface-bound probes are more accessible to target biomolecules and nanostructuring leads to greater surface areas [52]. Many efforts have been devoted to engineering spiky gold shells on various core particles [53,54,55,56,57].

We recently showed that growing gold spikes on magnetic particles allows for the combined separation and detection of oligonucleotides without the use of an extrinsic tag or secondary hybridization step [58]. In this work, we intended to go one step further in investigating and mastering the experimental parameters allowing for the synthesis of reliable stable spiky gold-coated superparamagnetic particles exhibiting fast magnetic separation and SERS effect. We used a magnetite cluster core coated with a silica shell that provides protection to the core from oxidation and offers a suitable surface for spiky gold shell formation. Prior to the nanostructure growth, we added gold seeds to the silica-coated magnetic nanoparticles, modified beforehand to exhibit four different surface groups, then we investigated the effect of surface chemistry on seed attachment and on further spike growth. We also explored the impact of different gold shell morphologies, spiky and bumpy, achieved by varying the growth conditions, on SERS enhancements and on particle stability over time.

## 2. Materials and Methods 

### 2.1. Materials

Gold(III) chloride hydrate (HAuCl_4_∙xH_2_O), cetyl trimethylammonium bromide (CTAB), sodium borohydride (NaBH_4_), silver nitrate (AgNO_3_), ascorbic acid, FeCl_3_·6H_2_O, sodium citrate dihydrate, polyacrylamide, tetraethoxysilane (TEOS), (3-aminopropyl)triethoxysilane (APTES), mercaptoundecanoic acid (MUA), (3-mercaptopropyl)trimethoxysilane (MPTMS), *N*-hydroxysuccinimide (NHS), and *N*-ethyl-*N*’-(3-(dimethylamino)propyl)carbodiimide hydrochloride (EDC), were purchased from Sigma Aldrich, (Paris, France). Millipore water and reagent grade ethanol (EtOH) were used.

### 2.2. Methods

#### 2.2.1. Particle Synthesis

Particle synthesis was done in several steps: magnetite sphere synthesis, silica coating and functionalization, gold seed binding, then growth of the gold seeds into spikes.

Magnetite spheres were synthesized using a previously developed hydrothermal synthesis method [34]. In brief, sodium citrate dihydrate, polyacrylamide, and FeCl_3_·6H_2_O were mixed and dissolved in Millipore water. A small amount of ammonium hydroxide was then added to the solution under vigorous stirring. This mixture was poured into a 125 mL PTFE-lined stainless steel pressure vessel (Parr) and heated at 200 °C for 12 h. The product was recovered magnetically and washed with deionized water, then by ethanol by magnetic decantation, and finally dried under nitrogen. Silica coating was performed in a second step. Briefly, the magnetite particle powder was dispersed into a solution of EtOH and Millipore deionized water by probe sonication. Ammonium hydroxide was then added to the dispersion, followed by the slow dropwise addition of TEOS in EtOH solution over 1 h under vigorous mechanical stirring. This mixture was then stirred at room temperature for 18 h, after which the product was recovered magnetically and washed with EtOH by magnetic decantation, then dried under nitrogen. Magnetic separation of samples in all steps was performed using either a small neodymium magnet or a rare earth homogenous magnetic separator (Sepmag Lab 2142, inner bore diameter 31 mm, radial magnetic field gradient 45 T/m). The samples were placed in or near the magnet and left to separate for 1–2 min. The supernatant was then gently removed using a pipette.

#### 2.2.2. Surface Functionalization of Silica-coated Particles

Amine functionalization: Silica-coated magnetite spheres were dispersed in 2:1 ethanol/ultrapure water for a final concentration of 5 mg/mL by bath sonication. Separately, a 20% *v/v* solution of APTES in Millipore water was prepared. While mechanically stirring the particles in a 50 °C water bath, APTES solution was quickly added so that the final APTES concentration was 2% *v/v*. After 24 h of stirring at 50 °C, the particles in solution were magnetically separated and decanted, then washed three times in Millipore water by magnetic separation and dried under a stream of nitrogen.

Short-chain thiol functionalization: Silica-coated magnetite spheres were dispersed in 1:1 toluene:ethanol for a final concentration of 5 mg/mL by bath sonication. After heating the solution to 50 °C, MPTMS was added to a final concentration of 2.5% *v/v*. The particles in solution were sealed in a vial and placed in a hybridization oven at 50 °C with gentle mixing. After 24 h, the particles in solution were magnetically separated and decanted, then washed once in toluene/ethanol and twice in ethanol by magnetic separation and dried under a stream of nitrogen.

Mixed amine/long-chain thiol functionalization: Amine-functionalized silica-coated magnetite spheres were dispersed in ethanol at 1 mg/mL by bath sonication. To activate the amine groups, 12 mM of NHS and 12 mM of EDC were added to the particles in ethanol and placed on a shaker for 90 min. The particles were magnetically decanted, redispersed in 10 mM MUA in ethanol, and placed on a shaker again for 90 min. The particles were then magnetically decanted and washed three times in ethanol by magnetic separation.

#### 2.2.3. Gold Growth on Functionalized Silica Surfaces

Gold seeds attachment: Silica-coated magnetite spheres were dispersed in ethanol at 2 mg/mL using a sonic bath for 20 min. Gold seeds were prepared by warming 5 mL of 0.2 M CTAB in Millipore water in a glass vial to 30 °C using a water bath, then by adding 0.125 mL of HAuCl_4_ to the vial while magnetically stirring. The bright yellowish-orange solution was stirred for 5 min. While still under magnetic stirring in the water bath, 0.3 mL of 0.01 M NaBH_4_ in Millipore water was added. The light brown solution was further stirred for 10 min. 

The silica-coated magnetite spheres, dispersed in ethanol, were combined with equal parts Millipore water and gold seeds (1 mL of each) and mixed by gentle shaking. The mixture was placed on a gentle rotating mixer for 1 h. The particles were then magnetically decanted, washed three times using magnetic separation in 1 mM CTAB, and redispersed in the same volume of 1 mM CTAB as the ethanol that was first used to disperse particles at 2 mg/mL.

Gold growth on the seeds: Growth solution was prepared by adding 0.5 mL of 0.01 M HAuCl_4_ and 0.1 mL of 0.01 M AgNO_3_ to 10 mM or 100 mM CTAB, then partially reducing the metal salts using 0.08 mL of 0.1 M ascorbic acid (final concentration of 0.5 mM HAuCl_4_, 0.1 mM AgNO_3_, and 0.8 mM ascorbic acid). When multiple samples were being synthesized, growth solution volume was scaled up accordingly. The solution was warmed to 30 °C using a water bath and magnetically stirred for 10 min. Following this, seeded particles were added to 10 mL of growth solution and mixed by inversion. For the first set of experiments, 100 μL of seeded particles were added to 10 mL of growth solution, and for the second set, after optimization, 400 μL of seeded particles were added to 20 mL of growth solution. The mixture was kept at 30 °C using either a water bath (without magnetic stirrer) or oven for 30 min. If growth occurred, the bath changed from a light brown to light grey (bound gold nanoparticles), dark blue (free gold nanoparticles), or a color in between. The particles in the bath were magnetically decanted, washed three times using magnetic separation in 1 mM CTAB, and redispersed in a volume of 1 mM CTAB equal to the volume of seeded particles used for storage.

#### 2.2.4. Techniques

UV-Vis spectroscopy: UV-Vis spectra of different suspensions were recorded in a plastic cuvette with a Cary 50 spectrophotometer (Varian, Inc., Paris, France). Spectral analysis of colloidal suspensions was performed in the range 300–800 nm. Milli-Q H_2_O was used as the blank.

Electron microscopy: Transmission electron microscopy (TEM) images were obtained using a JEOL JEM-2100 plus LaB6 (JEOL, Tokyo, Japan) microscope with an acceleration voltage of 200 kV and an Orius 4 K CDD camera (Gatan, Inc., Pleasanton, CA, USA). The images were analyzed using ImageJ Software version 1.51j8, NIH, (Bethesda, MD, USA).

Magnetic properties characterization: Magnetization curves were acquired by a superconducting quantum interference device (SQUID) magnetometer (Quantum Design, San Diego, CA, USA) at 300 K using particles dried in air. Magnetic separation times were acquired using the Sepmag (Barcelona, Spain).

X-ray photoelectron spectroscopy measurements (XPS): XPS analysis were performed on a Scienta Omicron Argus X-ray photoelectron spectrometer (Taunusstein, Germany), using a monochromated AlK (h = 1486.6 eV) radiation source having a 300 W electron beam power. The emission of photoelectrons from the sample was analyzed under ultra-high vacuum conditions (2 × 10^−10^ Torr). XPS spectra were collected in a fixed analyser transmission mode at pass energy of 100 eV for the survey spectrum and 20 eV for O_1s_, N_1s_, C_1s_, S_2s_, Si_2s_, and Au_4f_ core XPS levels. After data collection, the peak areas were determined after subtraction of a Shirley background. The atomic ratio calculations were performed after normalization using Scofield factors. Spectrum processing was carried out using the Casa XPS software package version 2.3.15 (www.casaxps.com).

Surface enhanced Raman scattering (SERS): Spectra were recorded in the 500-3400 cm^−1^ range on a modular Raman spectrometer, Model HL5R of Kaiser Optical Systems, Inc. (Ann Arbor, MI, USA) equipped with a high-powered near-IR laser diode working at 785 nm. Before spectra acquisition, an optical microscope was used to focus the laser beam into the MPym solution containing particles. The laser output power was 100 mW. For each spectrum, 10 acquisitions of 30 s were recorded to improve the signal-to-noise ratio. A detailed description of the experimental conditions adopted for SERS measurement is given in the SI section.

## 3. Results and Discussion

We grew gold spikes on silica-coated iron oxide spheres using a two-step method as depicted in Figure 1. First, CTAB-stabilized gold nanoparticle seeds were bound to the silica surface. Then, the seeds were grown into spiky particles using an aqueous CTAB-based growth solution. In a first stage, we investigated the influence of silica-iron core functionalization on gold seeds attachment and the further growth of gold spikes, then, when we identified the optimal surface chemistry, we explored, for the selected system, the impact of the experimental conditions applied for the growth phase on the gold nanostructures.

### 3.1. Silica-Iron Oxide Core Functionalization

We looked at growth on four differently functionalized silica surfaces to determine how the interaction between the surface and gold nanoparticles affects spiky shell growth. The four surface chemistries are shown in Figure 1. First bare silica, referred to as “bare”, then APTES modified silica leading to amine-terminated surfaces (NH_2_), MPTMS-modified silica forming thiol-terminated (SH), and finally a mixed layer exhibiting long chain thiols and amine functions (referred to as SH-NH_2_) obtained by MUA reaction on APTES-modified silica.

The particles were thoroughly analyzed with XPS to confirm successful functionalization and relatively estimate the functional group coverage. The XPS spectra of N_1s_, C_1s_, and S_2s_ are shown in Figure 2. The elemental composition and relevant atomic ratio are shown in Table 1.

The survey spectra showed that all of the expected elements were present on the particles. Nitrogen groups were present on both NH_2_ and SH-NH_2_ terminated surfaces in a similar amount, which was significantly larger than the trace signal of nitrogen on bare particles (likely due to the ammonium hydroxide used in synthesis). Carbon contamination was also present on bare particles but the C_1s_ signal increased largely when APTES was grafted and further increased upon MUA reaction on APTES, confirming the successful silanization of the particles. For SH surfaces, obtained by MPTMS reaction, the change in C_1s_ signal was barely noticeable, showing a modest grafting of this silane on the surface. The S_2s_ peak was present as shown in Figure 2 and Table 1, attesting for MPTMS reaction, yet, its atomic percentage (Table 1) was much lower than that measured for nitrogen upon APTES reaction. SH coverage barely reaches 10% of the NH_2_ one. A similar result was previously observed on planar silica surfaces [59] and can be easily explained by the catalytic role of the amine group of APTES in the hydrolysis-condensation process that favors its grafting and leads to a high density of silanes on the surface. Following the MUA reaction on NH_2_ particles, the S_2s_ signal appeared confirming its successful grafting. The amount of bound MUA was estimated to be around 13% of the amine groups, which is lower than the results previously observed on planar surfaces (approx. 24%). Nevertheless, it resulted in a thiol density comparable to or even slightly higher than that on the short-chain SH particles.

### 3.2. Gold Seeds Attachment

Small CTAB-stabilized gold nanoparticles were utilized as seeds for further growth of gold spikes. The silica-iron oxide core particles with the four surface chemistries described above (Figure 1) were impregnated by these seeds, then the unbound nanoparticles were removed by multiple washing and redispersion steps. Gold seed binding to particles was characterized by both TEM and UV-Vis; the resulting images and spectra are shown in Figure 3 and Appendix A.

UV-Visible absorption spectra for AuNPs show a typical LSPR band at 530 nm while the bare silica-iron oxide core particles exhibited a broad absorption in the visible region. After impregnating the particles by AuNP seeds, TEM images showed that the size and number of gold seeds varied with type of surface functionalization. Larger AuNPs (3.5–5 nm) were seen on bare (Figure 3A) and on SH functionalized (Figure 3C) silica surfaces while smaller seeds (<1 nm) were seen on NH_2_-SH (Figure 3D) silica surfaces. Almost no gold nanoparticles were seen on NH_2_ silica surfaces (Figure 3B). It should also be noted that smaller particles (<1–2 nm) might not be observable by TEM due to the resolution limit. UV-Vis spectra shown in Appendix A are also informative with respect to seed attachment, despite the original absorption of particles dominating the spectra. Indeed, after gold seed binding, the spectra show a red-shift towards AuNPs LSPR band, except for NH_2_ terminated surfaces, where, in confirmation with the lack of AuNPs bound seen in TEM, the UV-Vis spectrum showed very little shift.

The first hypothesis to explain these differences observed for seeds binding would be electrostatic. The CTAB-stabilized nanoparticles are positively charged, therefore a strong interaction will occur with silica surface which PZC (point of zero charge) is around 2 and silica surface strongly adsorbs positively charges species above this value [60,61]. Amine terminated surfaces, on the other hand, are positively charged, which inhibits AuNP seed adsorption and possible interactions between nitrogen atoms and the gold surface. For SH terminated surfaces, the MPTMS coverage is low and the electrostatic adsorption on bare silica is likely favored even if covalent attachment to the thiol groups should occur through the S-Au affinity. The most interesting surface chemistry is NH_2_-SH, where the long thiol chains seem to overcome the barrier of electrostatic repulsion with amine groups. The small size of the seeds compared to that observed for SH terminated surface is possibly the result of the absence of electrostatic interaction with the underlying layer and the anchoring of the seeds solely to SH groups.

### 3.3. Spike Growth on Seeded Particles

A growth solution containing gold salt, CTAB, and silver nitrate was combined with the seeded particles and the gold structures were left to grow. These were then analyzed by TEM (Figure 4) and UV-Vis spectroscopy (Appendix A).

Based on TEM images of Figure 4, spiky gold shells grew on all silica surfaces except for the amine-coated surface (Figure 4B), where only gold stars can be seen on the surface of the particles. Shell coverage was greatest on the long-chain functionalized SH-NH_2_ particle surfaces (Figure 4D) and was at an intermediate amount on the bare (Figure 4A) and short-chain SH (Figure 4C) functionalized particles.

As would be expected based on the method of synthesis, shell growth results from the individual growth of gold seeds into anisotropic gold shapes (nanostars, bipyramids, etc.), which eventually coalesce into full shells surrounding the particles. UV-Vis results shown in Appendix A. Appendix A may seem poorly informative due to the broad absorption of the particles but they do correlate well with TEM data. Indeed, several researchers have studied the optical response of gold shells and have shown that, as individual seeds begin to grow and coalesce, intercoupling between particles leads, first, to a broadening of the plasmon peak and shift to longer wavelengths, then, to a shift to shorter wavelengths as the shell approaches full coverage [46,62]. The results show that plasmon peak broadening and red shift occurs upon gold growth on particles with bare or SH surfaces—those that came closer to approaching full shell growth—while very little shift occurred upon gold growth on NH_2_ surfaces. In addition, on the long-chain functionalized SH-NH_2_ surfaces, we see the appearance of a peak at around 630 nm along with the increase of a broad peak reaching into the NIR range, previously ascribed, based on the experimental results and simulations, to changes that occur upon complete shell formation [46,62].

Differences in spiky shell growth can be seen as a direct result of the differences in gold seed binding to the surface. More gold seeds bound to SH surfaces than to bare or NH_2_ surfaces because of the previous discussion as well as the strong gold-thiol bond that may form. Few gold seeds bound to the amine-functionalized surface because of electrostatic repulsion, then further seed growth enhanced this low binding since growth involves CTAB as a stabilizing group as well. Some gold nanostructures are still seen around the amine particles, possibly because of the strong gold-amine bond that may form, despite electrostatic repulsion between the surface and stabilizing groups.

### 3.4. Magnetic and SERS Properties of the Spiky Nanoparticles

The idea behind using a magnetic core is that particles can be easily separated from a solution using a magnetic field. Use of the particles in practical applications, such as the separation of biomolecules, requires fast separation times and the ability to be redispersed before and after separation. The magnetic properties of the particles were studied before and after gold coating to determine whether or not the particles fit these requirements. Note that, for all steps in the synthesis, magnetic separation was used to wash the particles, and, during these steps, the particles were easily dispersed in solution and quickly separated out under the influence of a magnet which clearly indicated that they retained their magnetic properties. A SQUID magnetometer was used to obtain magnetization curves of the particles before and after gold coating (Appendix A). Gold shell growth on the silica-coated iron oxide particles lowered the saturation magnetization $(M_{s})\;$ from 32 to 9 emu/g and after gold shell coating, suggesting that some reduction in magnetization occurs upon coating, as previously observed by other authors [63,64]. In spite of this diminution, both types of particles separated quickly, with opacities reaching below 1% in less than two min (Appendix A and Appendix A).

To study whether or not the nanostructures prepared above demonstrate Raman scattering enhancement, we used 2-mercaptopyrimidine (MPym) as a Raman probe molecule because of its high Raman activity and ability to strongly interact with the gold surface through S-Au bonds. The Raman spectra of 2-mercaptopyrimidine, at a concentration of 10 μM, are shown in Figure 5A for the four spiky particles. Figure 5A also shows the Raman spectrum of MPym without particles at a concentration of 10 mM, i.e., 10^3^ higher than with the particles, added to demonstrate the enhancement by particles compared with the unenhanced signal. 

All the particles with spiky gold shells demonstrated SERS enhancement on the order of 10^4^, a value comparable with those seen in literature [26]. We also measured Raman spectra before gold growth using only the seeded particles bound to silica-coated magnetite particles in combination with 10 μM MPym and no signal was observed, indicating no enhancement. 

We also checked whether the Raman signal varied with MPym concentration. Figure 5B shows the spectra obtained for increasing MPym concentrations in the presence of particles with spiky gold shells on bare silica-coated magnetite cores. In the absence of MPym, all the observed bands (labelled in black on the spectra) are ascribable to CTAB, whose signal is enhanced by the nanoparticles. The main band at 756 cm^−1^ corresponds the symmetric stretch of the trimethylammonium head group from, at 1441 cm^−1^ are observed the CH_2_ scissor modes and the amine-bound C-H symmetrical deformation, and finally, the bands in the 2840–3000 cm^−1^ are due to symmetric and antisymmetric stretching vibrations modes of CH_2_, terminal CH_3_, and N-CH_3_ [65,66].

The signal from bands corresponding to MPym increased progressively with increasing concentrations. The main bands present are the following: at 990 cm^−1^ ring breathing modes, at 1073 cm^−1^ in-plane stretching modes, at 1156 and 1375 cm^−1^ CH bending, and at 1539 and 1562 cm^−1^ ring stretching [67]. The lowest concentration of MPym clearly detectable on the spectra was less than 100 nM. Above this concentration, the MPym bands dominated the spectra while the CTAB signal decreased. To quantitatively estimate this trend, we plotted the height of the band at 1073 cm^−1^ as well as that of CTAB characteristic bands as a function of MPym concentrations (Appendix A). As would be expected from a surface sensitive method, the signal increase showed saturation behavior, as demonstrated by a fit to a Langmuir model. All the same, CTAB bands decreased gradually suggesting that the MPym can successfully replace CTAB groups at the surface. It is important to note, though, that with CTAB groups still present on the surface, the observed limit of detection of MPym is likely hindered by the presence of these groups. Future applications will require ensuring that the stabilizing groups do not interfere with detection limits.

One additional important observation is that particles with gold shells on differently functionalized silica surfaces produced different signal intensities. The largest intensity was seen using particles with gold shells formed on bare silica-coated magnetite cores, and was approximately four times larger than the intensity seen using particles made with amine or long-chain SH-NH_2_ surfaces and 2.6 times larger than the intensity seen using particles made with short-chain SH surfaces. The reason for the differences is thought to be due to an optimal amount of gold coverage, i.e., too little gold and the surface available for enhancement is small, but too much gold and the particle suspension becomes unstable, leading to particle aggregation. Aggregation was observed in the case of SH-NH_2_ particles, and to a more limited extent, SH particles. NH_2_ particles showed less gold coverage than the other particles. Another possibility that could explain the differences, still due to an optimal amount of gold coverage, is that an incomplete nanoshell shows greater enhancement than a complete one [46], possibly due to an optimal size and number of nanogaps between surface-bound gold nanostructures. Despite this, the long-chain SH-NH_2_ particles resulted in the most gold coverage and the least amount of gold nanoparticles observed free in the supernatant. Based on these observations, we concluded that these particles had the greatest potential for reproducible measurements and longer-term stability. All following experiments were therefore performed using long-chain SH-NH_2_ particles, but using 10 mM CTAB instead of 1 mM CTAB for the washing steps. This minor modification prevented aggregation and resulted in stable particles with sufficiently high SERS signals. Additionally, in what follows CTAB was used as Raman reporter rather than MPym, to prevent any change in particles dispersion that would result from thiol reaction with gold surfaces.

### 3.5. Impact of the Growth Parameters on Nanoparticles’ Shape: Spiky vs Bumpy Nanoparticles

We wanted to investigate how changing the growth conditions impacts the morphology of gold nanostructures and their SERS enhancement. Many parameters can affect gold growth in CTAB-based syntheses [68], including growth time [54,69], reagent concentrations [69,70,71,72,73], and the presence of other halide ions [69,71]. In what follows, we investigated two important ones: the influence of growth time and CTAB concentration.

Figure 6 shows the TEM images and corresponding UV-Vis spectra for growth time of 2, 15, and 30 min and CTAB concentrations of 10 and 100 mM. As expected, increasing the time spent in the growth bath increased the amount of gold bound to the surface. All the same, increasing the concentration of CTAB in the growth bath decreased the amount of gold bound to the surface at a given time.

In addition to the amount of bound gold, CTAB concentration also impacted the morphology of particles. Indeed, with a lower CTAB concentration (10 mM) the shells were less spiky and exhibited lower aspect ratio protrusions resulting in rather bumpy than spiky particles. Growth also occurred more quickly when a lower CTAB concentration was used but the time parameter did not affect the morphology of particles. We also tried an even lower CTAB concentration, 1 mM, (results not shown) that resulted in a spontaneous HAuCl_4_ reduction to Au_0_ by ascorbic acid, forming AuNPs before adding the seeded-particles. The critical micelle concentration (CMC) of CTAB is approximately 1 mM, so, a possible interpretation of these data is that gold chloride ions in the bath solution are stabilized by micelles as long as the concentration of CTAB is above the CMC [74]. Since ascorbic acid is a weak reducing agent, gold seeds are required as nucleation sites for complete reduction of the gold salt to occur. 

For both bumpy (10 mM CTAB) and spiky (100 mM CTAB) particles, the UV-Vis absorbance increased in the NIR region with increased growth time. This result fits with previous research showing that hybridization between the resonances of the individual gold nanostructures on the particle surface leads to a broad NIR absorption band that red shifts with increased coverage [46,71]. In what follows, we discuss the data obtained after 30 min growth and using 100 mM CTAB for both spiky and bumpy nanoparticles. 

#### 3.5.1. Elemental Composition of Spiky and Bumpy Nanoparticles

Prior to SERS enhancement investigation, we assessed the elemental composition of the spiky and bumpy particles with the aim of determining the precise content for each element. This information is needed to establish whether the differences in SERS enhancement result from different amounts of gold, silver, or CTAB bound to the surfaces, or whether the morphology did in fact play a role. The atomic percentages for Si_2p_, O_1s_, Au_4f_, Br_3d_, and Ag_3d_ for spiky and bumpy nanoparticles are gathered in Table 2, along with some relevant elemental ratios.

As suggested by TEM images, more gold is reduced on bumpy particles than on spiky ones, and this is confirmed by both the atomic content in Au and the attenuation of Si and O coming from the underneath silica. Considering that XPS is a surface technique, calculating the relative contents based on the ratio to silicon is a more accurate way to estimate the elemental content of each atom as, in this case, the attenuation of the elements by the adsorbed layers is taken into consideration. 

When comparing the Au/Si ratio, it appears that bumpy particles accommodate three times more gold than the spiky ones. The same factor 3 is observed for silver and for bromine signal due to CTAB. The ratio Br/Au was therefore the same for both particles, i.e., 0.13. This value is not significant in terms of surface chemistry since both adsorbed and unbound CTAB can be present in the sample, however, it is relevant when it comes to comparing SERS CTAB signal on both particles. To summarize XPS data, bumpy particles contain three times more gold, silver and CTAB. Therefore, if SERS enhancement was to be dependent on the amount of substance in the samples, it should be three times higher for bumpy compared to spiky.

#### 3.5.2. SERS Enhancement of Spiky and Bumpy Nanoparticles

We followed the CTAB signal to investigate how SERS enhancement varied across the two morphologies. TEM images of the particles and the Raman spectra taken of the particles in a solution of 167 µM CTAB along with the Raman spectra of unenhanced 0.2 M CTAB (for comparison) are shown in Figure 7.

Spiky particles led to greater SERS enhancement than the bumpy ones. Using the height of the peak at 763 cm^−1^ (corresponding to the trimethylammonium headgroup [65,66]) to compare the enhancement shows that the enhancement due to the spiky particles was 1.4 times greater than that due to the bumpy particles. Comparing the signal to the CTAB signal has some practical issues—namely that the SERS selection rules for the modes observed and their relative strengths are different than for the standard SERS—but the analytical enhancement factor (AEF) appears to be on the order of 10^4^ for both types of nanostructured particles. This value was determined according to the expression $AEF=\frac{I_{SERS}/c_{SERS}}{I_{RS}/c_{RS}}$, where *I_SERS_* and *I_RS_* are the enhanced and unenhanced Raman signals, *c_RS_* is the concentration of molecules in the unenhanced bulk sample, and *c_SERS_* is the concentration of molecules in the SERS sample [75].

Bumpy particles contain higher amount of substance for Au and Ag, 3 times higher as evidenced by a set of elements in XPS, yet the enhancement is higher for spiky particles. We can therefore claim that the enhancement is solely related to the morphology. Spiky particles morphology exhibits sharp tips, more protrusions and nanostructures, and consequently higher surface to volume ratio, which may impact the enhancement either through surface effects, i.e., greater surface area available for CTAB adsorption, or through electromagnetic effects due to the sharp tips and hot spots. The surface effect is not validated here as the ratio Br/Au was identical for both spiky and bumpy. Therefore, we can conclude here that the larger Raman enhancement on the spiky shell particles is due to increased electromagnetic enhancement caused by the spike shape.

#### 3.5.3. Stability Overtime of Spiky and Bumpy Nanoparticles

Using the particles in practical situations requires that they are stable over a period of time. We synthesized spiky and bumpy particles and measured their absorbance (UV-Vis), change in morphology (TEM), and SERS enhancement (Raman spectroscopy) after 1 week and 1 month. Particles were washed three times in 10 mM CTAB before each measurement. The TEM results together with UV-Vis and Raman spectra are shown in Figure 8.

The particles remained fairly stable after one week, although some morphological changes did occur. After one month, the UV-Vis absorbance of the particles decreases overall, but especially large decreases occur in the NIR range. We can observe a resemblance between these curves and those seen after short growth times (see Figure 6), as less complete shells show lower NIR absorbance because of decreased plasmon hybridization. The NIR absorbance decrease of the spiky particles is also likely due to the reduced aspect ratio of the gold nanostructures, as seen in TEM.

After one week, the bumpy particles remained stable while the spiky particles showed a change in SERS signal. Surprisingly, the SERS signals increased over time, with both spiky and bumpy particles showing the strongest signals after particles had been left for one month.

The morphological changes that occur over time are likely the result of Ostwald ripening, where the “spiky” areas with a small radius are less energetically stable and over time, the atoms migrate to areas with a larger radius that are more energetically stable. This phenomenon was previously observed experimentally and theoretically when monitoring the long-term stability of CTAB-stabilized gold nanostars [76]. To overcome these changes over time, one possible solution is replacing CTAB by another stabilizing group, either through grafting of thiolated molecules, such as 3-mercaptopropionic acid, that will form monolayers through Au-S bonds and have proven to be efficient in greatly improving the stability of high aspect ratio nanostars [77], or by silica coating the spiky particles, provided that the layer is thin enough to allow for the SERS enhancement, which has also proven to be efficient in maintaining the morphology of spiky nanostars [78,79,80].

## 4. Conclusions

We synthesized spiky gold-coated superparamagnetic particles that are magnetically separable and can be used for surface enhanced Raman scattering (SERS) detection of model molecules in a concentration-dependent manner. These nanostructured objects were prepared starting from silica coated magnetite particles that were functionalized to attach gold seeds, then grown into spiky structures. By controlling the surface chemistry applied to attach the seeds, we obtained different geometries and gold coverage. We also varied the growth conditions to engineer either spiky or bumpy particles. The morphology of the nanostructured shell plays a role in SERS enhancement, i.e., spiky shells led to a greater enhancement than bumpy shells, but the morphology also affects the stability of the particles over time, as bumpy shells are more stable than spiky shells, showing fewer morphology changes and less change in SERS enhancement over time. Our findings demonstrate a first step in the development of particles that can be used for combined separation and detection of biomolecules. Ongoing work on the further stabilization of spiky particles via a thin silica layer coating aims to overcome the stability issues and expand the applications of these nanomaterials.

## Figures and Tables

**Figure 1 nanomaterials-10-02136-f001:**
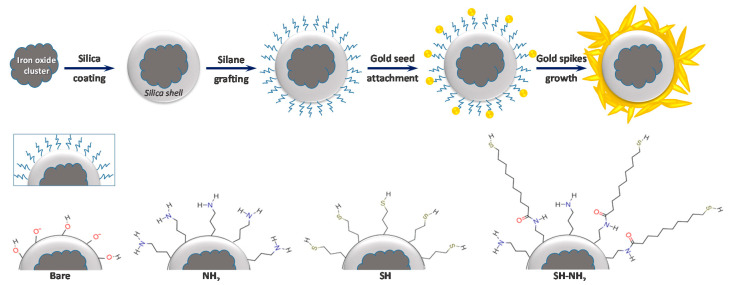
Strategy adopted for the synthesis of spiky particle starting from magnetite sphere, gold seed growth was tested on four surface termination: bare silica, amine-terminated (NH_2_), short thiol (SH), and mixed amine-long thiols (SH-NH_2_).

**Figure 2 nanomaterials-10-02136-f002:**
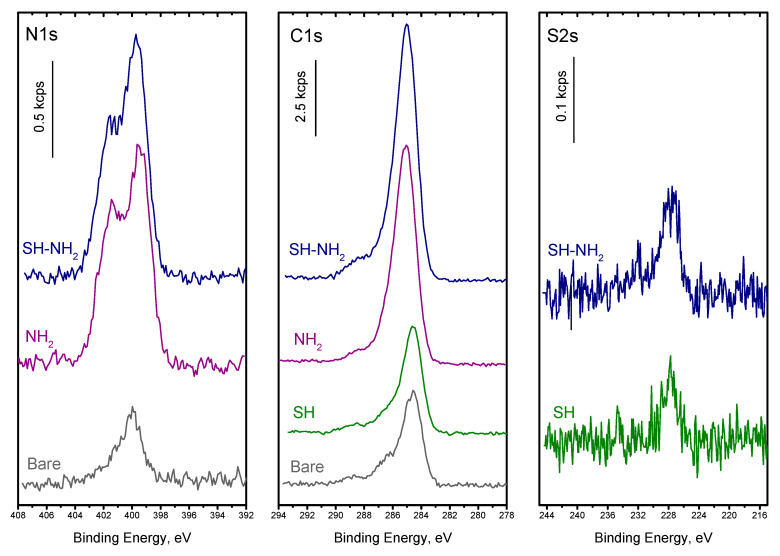
XPS data for N_1s_, C_1s_, and S_2s_ peaks recorded for bare silica, SH, NH_2_, SH-NH_2_ surface chemistries.

**Figure 3 nanomaterials-10-02136-f003:**
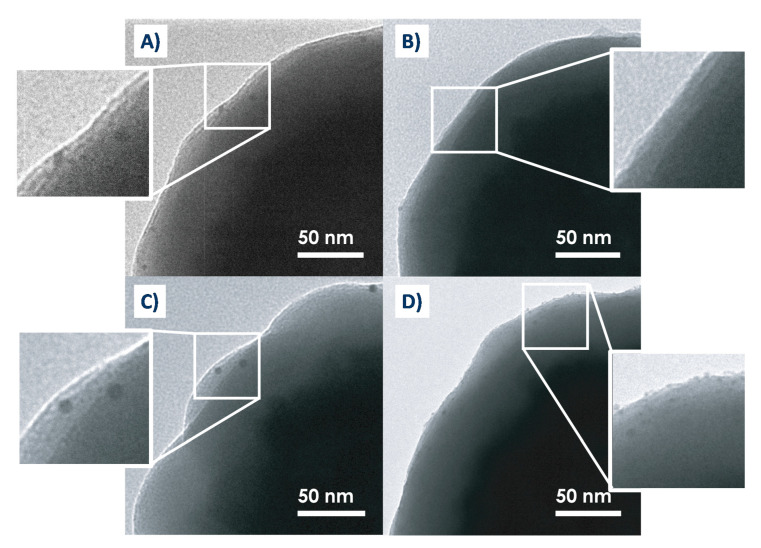
TEM images of particles after gold seed binding on (**A**) bare, (**B**) NH_2_, (**C**) SH, and (**D**) SH-NH_2_.

**Figure 4 nanomaterials-10-02136-f004:**
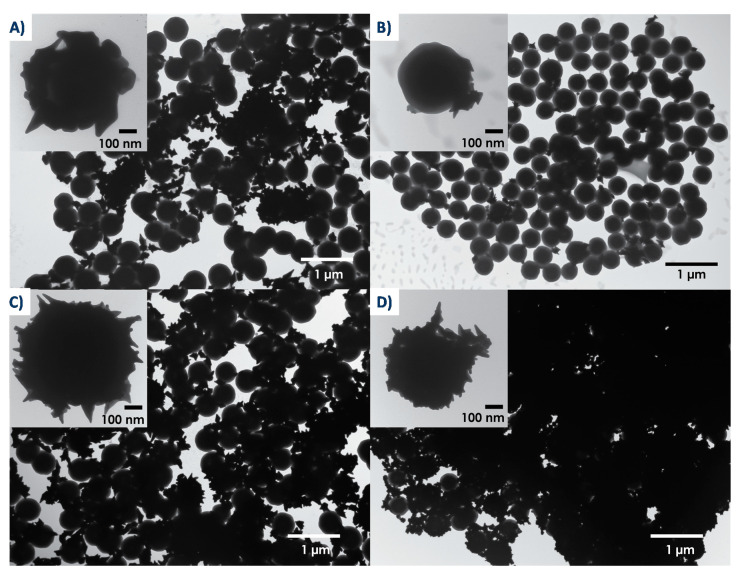
TEM images of spiky particles, with spiky nanostructures grown on (**A**) bare, (**B**) NH_2_, (**C**) SH, and (**D**) SH-NH_2_. Insets show high magnification of spiky shells.

**Figure 5 nanomaterials-10-02136-f005:**
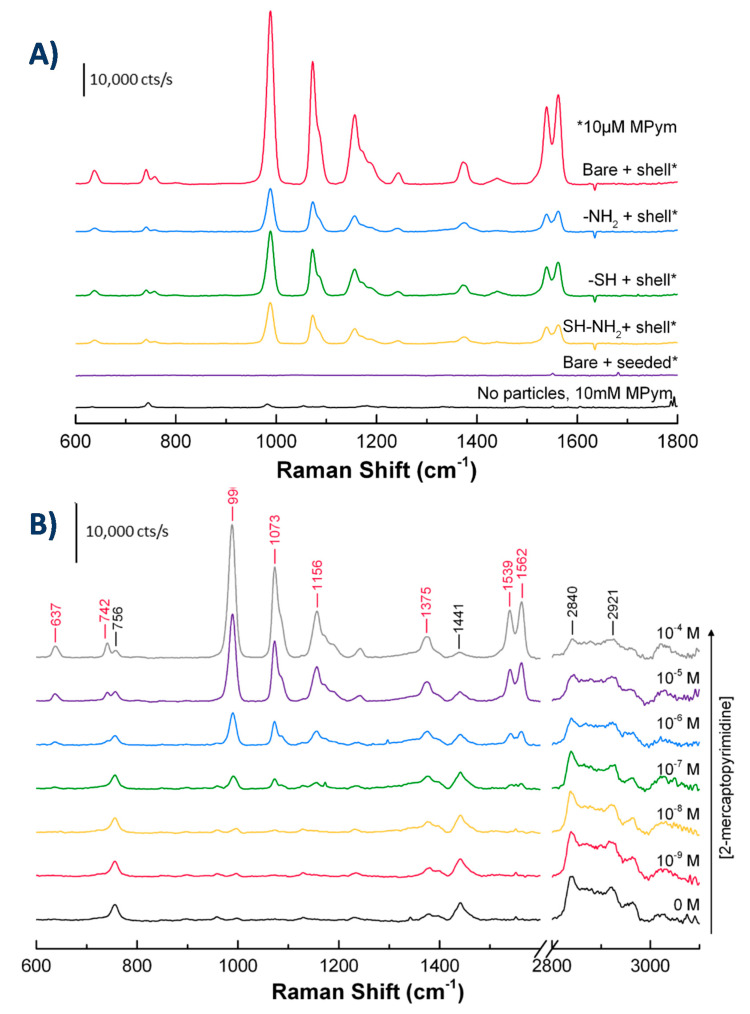
(**A**) Raman spectra of 2-mercaptopyrimidine (MPym) without (10 mM) and with particles (10 μM). (**B**) Raman spectra of MPym at different concentrations with particles featuring spiky gold shells on bare silica-coated magnetite particles, MPym peaks are labelled in red and CTAB peaks in black. All samples used 50 μL of particles in 1 mM CTAB added to 3 mL of water.

**Figure 6 nanomaterials-10-02136-f006:**
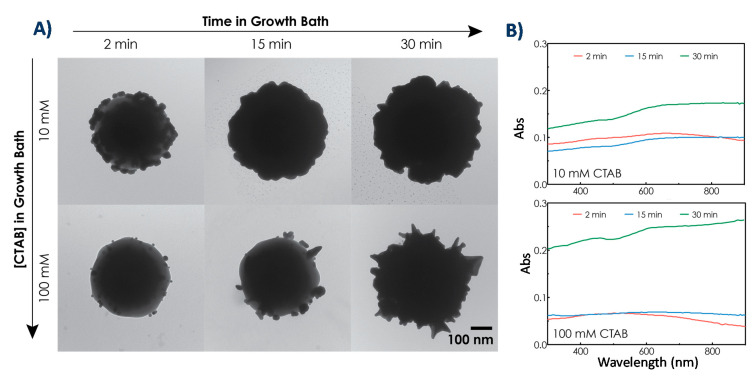
(**A**) TEM images showing representative particles at varying growth times and CTAB concentrations and (**B**) corresponding UV-Vis extinction spectra for particles synthesized at different growth times using (top) 10 mM and (bottom) 100 mM CTAB.

**Figure 7 nanomaterials-10-02136-f007:**
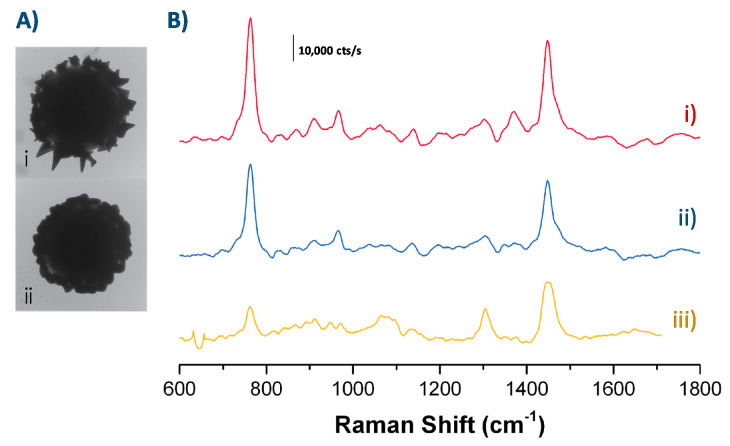
(**A**) TEM images and (**B**) corresponding Raman spectra of particles in 167 µM CTAB with (i) spiky shells, (ii) bumpy shells, and (iii) Raman spectra of 0.2 M CTAB in water (1200× more concentrated than particle-enhanced measurements).

**Figure 8 nanomaterials-10-02136-f008:**
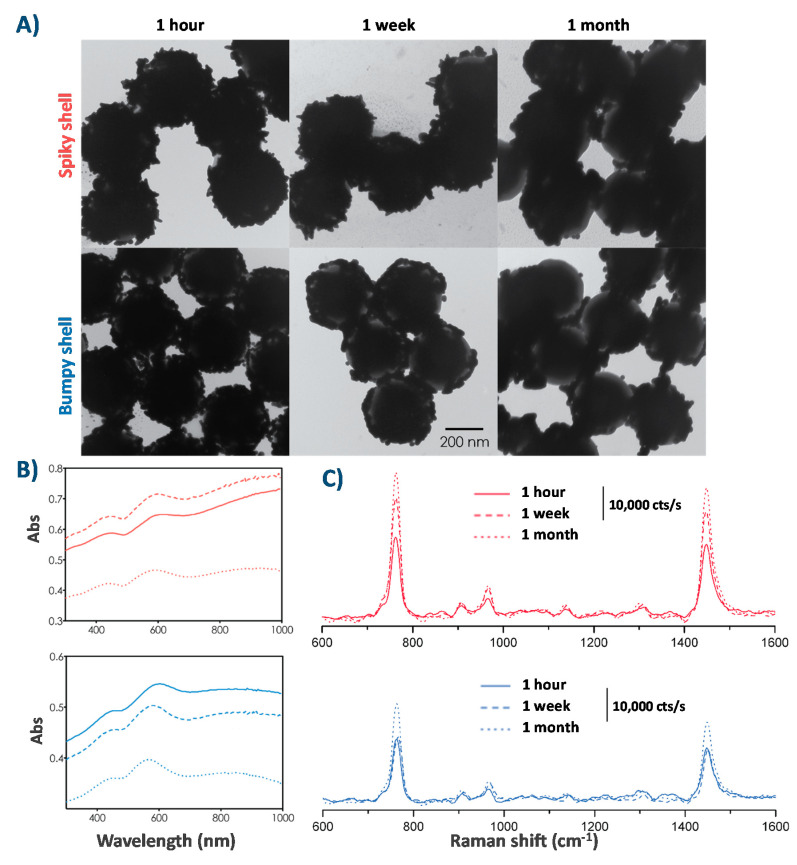
(**A**) TEM images showing representative particles over time, (**B**) and (**C**) corresponding UV-Vis and Raman (in 167 μM CTAB) spectra for spiky (top) and bumpy (bottom) particles over time.

**Table 1 nanomaterials-10-02136-t001:** Elemental Percentages from XPS Results for the Four Surface Chemistries and Relevant Ratios.

Surface	Si_2p_	O_1s_	C_1s_	N_1s_	S_2s_	C/Si	O/Si	N/Si × 100	S/Si × 100
Bare	33.8	56.6	9.0	0.59	-	0.27	1.62	1.7	-
SH	33.6	55.4	10.7	-	0.28	0.32	1.65	-	0.83
NH_2_	30.1	46.0	21.3	2.59	-	0.71	1.53	8.6	-
SH-NH_2_	27.5	44.4	25.1	2.63	0.34	0.91	1.62	9.6	1.24

**Table 2 nanomaterials-10-02136-t002:** Elemental Percentages from XPS Results and Relevant Ratio for Spiky and Bumpy Particles.

Shape	Si_2p_	O_1s_	Au_4f_	Ag_3d_	Br_3d_	Au/Si	Ag/Si	Br/Si	Br/Au
Spiky	15.9	22.2	7.7	1.7	1.0	0.48	0.11	0.06	0.13
Bumpy	7.6	14.1	10.8	2.4	1.3	1.42	0.31	0.18	0.13

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
