# Peer review of "Nanostructured and Spiky Gold Shell Growth on Magnetic Particles for SERS Applications"

_nanomaterials, 2020, doi:10.3390/nano10112136_

Round 1

Reviewer 1 Report

The authors demonstrate the use of spiky gold nanoparticles with iron oxide cores for SERS application. They used different surface chemistries to modify the growth of gold spikes. My concerns are listed below.  

  1. Do the particles retain the magnetic properties after gold spike formation, it will be a good idea to check the surface morphology and absorption spectra before and after exposing them to a magnetic field. Further, can the particles be separated by a magnetic field? 
  2. In Figure 3A, the authors should look at more representative images of each condition with the TEM images showing the difference in the growth conditions. 
  3. In Figure 3B, the authors should consider understanding how the theoretical absorption spectra should change depending on the growth condition and if the experimentally obtained absorption spectra matches the theoretical prediction.
  4. Figure 3C and 3D are not clear enough to observe the spikes that the authors describe.
  5. Figure 5 should be summarized as a dose response curve with different surface chemistries in order to better see the difference in the SERS spectra between the conditions. 
  6. In Figure 6B, how does the theoretical absorption spectra look like in comparison with the experimentally obtained spectra? Are the particles aggregating?

Reviewer 2 Report

The manuscript describes the preparation and characterization of particles comprising a magnetic core covered with a spiky gold shell and investigates their use as substrates in surface enhanced Raman spectroscopy (SERS). Authors have already published in a previous manuscript the preparation of the magnetic core and the functionalization with the spiky gold shells using different linkers, which was applied to the separation and detection of oligonucleotides via SERS. Herein the effect of different linkers is evaluated, as well as the impact of the gold morphology, namely spiky or bumpy. The model analyte 2-mercaptopyrimidine is used, although there is no application to further analytes neither real samples. The topic is interesting; however, there are some issues that need to be addressed or clarified before the manuscript could be published.

The following suggestions are given for improving the manuscript:

1) Authors have prepared magnetic nanoparticles covered with the spiky gold shell for SERS applications, revealing the effect of the shape of the gold layer on the enhancement observed. However, very little information on the analytical procedure is provide, so that the role of magnetism of these nanoparticles is not justified at all in the sample treatment. The method has been evaluated just with standards. To which type of samples are these substrates intended to be used?

2) As previously mentioned, authors should provide more details about the sample preparation for SERS measurements. For example, for those in Figure 5. Were those measurements carried out in solution? Was the dispersion drop-casted and dried on a surface? Please give more details about how those measurements were performed.

3) Please include the units in brackets in all the axes from the graphs (e.g. y-axes fig. S1 and S3, Fig. 3., Fig. 6b, Figure 7b (also include what is measured in y-axis). The same applies to Figure 8.

4) In section 2.1. instead of the sentence “Silica-coated iron oxide particles were synthesized previously at the University of Waterloo as described in reference [35]”, the different reagents used in the synthesis of magnetite sphere and their coating with silica should be mentioned along with their suppliers.

5) Although of common use, all abbreviations should be described the first time they are used, e.g. page 3 EtOH. Also, please define PymSH in line 185 before use.

6) Please revise the spelling, for example the “L” of liters should be always written in capital letters, e.g. “mL” in line 153. Also line 177 XPS spectra. Line 182 “cm-1” revise superscript.

7) Please revise the sentence in lines 190-194. “[…] then, once the optimal surface chemistry was identified, we explored […]”.

8) I would recommend rather than using sentences such as “we have synthesized” or “we investigated…” the use of passive form, e.g. “nanoparticles were synthesized” or “the influence of variable was investigated”.

9) Authors should include in the supplementary file microscopy (SEM or TEM) images of the magnetic nanoparticles and the magnetic nanoparticles after silanization. Moreover, TEM images of the gold seeds before binding should be included, commenting their size and size distribution (i.e. those whose spectra is shown in grey in Fig. 3B).

10) In figure 3B, please write the abbreviation AuNPs with the “P” always in capital letters for homogenization.

11) Authors include TEM images of the particles after the growth of the spiky structures on the different functionalized magnetic nanoparticles. Would it be possible for authors to provide an EDX mapping of the structures, showing the difference in composition in the different points?

12) Information on the size and size distribution of the final spiky particles should be provided.

13) Authors should explain how the enhancement factors (EFs) observed in SERS were calculated the first time EFs are mentioned. It is mentioned later on page 13, but this should be moved to the first place where enhancement factors are discussed. The different EFs for the different types of spiky particles should be calculated and given.

14) Error bars should be added in Fig. S3.

15) Information about the reproducibility of the SERS measurements in term of relative standard deviation should be provided.

16) Why was CTAB used instead of 2-mercaptopyrimidine (MPym) for studying the SERS enhancement of spiky versus bumpy shells? Authors should explain and justify this selection.

Reviewer 3 Report

The manuscript deals with investigations of Fe3O4 silica coated nanoparticles (NPs) that are additionally coated with gold seeds, which depending on the surface functionalization lead to the formation of spiky nanostructures with various characteristics. Depending on the growth conditions, also bumpy gold nanostructures on the particles’ surface are obtained. The authors show the enhancement of the surface-enhanced Raman (SERS) intensity when the NPs are put in a 2-mercaptopyrimidine solution. The particle stability and SERS enhancement as a function of time and shape is also investigated. The authors conclude that the spiky gold nanostructures show larger enhancements, but due to the higher aspect ratio than in the bumpy ones, the NPs are less stable, leading to increased morphological changes.

The paper is well written. The results presented are complete, well presented and well founded by employing numerous techniques to do so. Their results are an extension of their previous work [B.E. Bedford et al. “Spiky gold shells on magnetic nanoparticles for DNA biosensors”, Talanta 182, 259-266 (2018)], where the authors actually show the applicability of their system.

I have just few suggestions for the authors concerning the format:

  • I believe for consistency, all subfigures in the Figures should be labelled throughout the entire manuscript either with capital or with non-capital letters: a) or A), but not a mixture of both.
  • The quality of the absorbance spectra could be improved.

Round 2

Reviewer 1 Report

The authors responded to my comments satisfactorily.

Author Response

We thank the reviewer for her/his positive feedback

Reviewer 2 Report

The manuscript evaluates the effect of different linkers on the SERS application of magnetic core nanoparticles covered with a spiky or bumpy gold shell. Authors have addressed most of the issues raised by the reviewers, thus improving the manuscript. Nevertheless, there are some issues that remain unanswered:

1) Regarding the error bars of Fig. S3. Authors claim they are drawn from Fig. 5B. How many measurements were carried out at the same concentration? For calibration purposes usually at least three independent measurements are carried out, thus the error bars provide information on the precision of the measurements.

2) Related to that, authors were requested to provide information of the reproducibility of the SERS measurements, however no quantitative information in terms of relative standard deviation (RSD) of different measurements is provided. Sentences such as “we concluded that these particles had the greatest potential for reproducible measurements and longer-term stability” should be accompanied by data supporting them.

Author Response

We have already responded to these two points addressed by the reviewer in her/his first report, and, with all due respect, the answers were already clearly indicated in the manuscript. In the last part of the manuscript we discuss SERS signal evolution with time and particle aggregation and we envision the solutions to overcome these issues. Unlike what is stated by the reviewer, our objective was not limited to studying “the effect of different linkers on the SERS application of magnetic core nanoparticles covered with a spiky or bumpy gold shell” we deeply investigated the experimental parameters allowing for the synthesis of reliable spiky gold-coated superparamagnetic particles exhibiting fast magnetic separation and SERS effect. Again, our work is not an analytical investigation.